# Gender Differences in Oesophageal Squamous Cell Carcinoma in a South African Tertiary Hospital

**DOI:** 10.3390/ijerph17197086

**Published:** 2020-09-28

**Authors:** Lucien Ferndale, Colleen Aldous, Richard Hift, Sandie Thomson

**Affiliations:** 1Department of Surgery, Greys Hospital, Pietermaritzburg 3200, KwaZulu Natal, South Africa; 2College of Health Sciences, School of Clinical Medicine, University of KwaZulu-Natal, Durban 4013, KwaZulu Natal, South Africa; Aldousc@ukzn.ac.za (C.A.); hift@ukzn.ac.za (R.H.); 3Division of Gastroenterology, Department of Medicine, University of Cape Town, Cape Town 7925, South Africa; sandie.thomson@uct.ac.za

**Keywords:** oesophageal squamous cell carcinoma, risk factor exposure 2, gender differences

## Abstract

(1) Oesophageal squamous cell carcinoma is common in Africa and has a male preponderance. The gender-based differences in clinical presentation and risk factor exposure are poorly studied in the African context. Our aim was to compare males and females with this disease. We analyzed the differences in clinical features and risk factor exposure between males and females with oesophageal cancer. (2) Data from patients presenting to a tertiary hospital in South Africa with oesophageal squamous cell carcinoma were analyzed. Data collected included patient demographics, clinical presentation, pathology and risk factor exposure. (3) Three hundred and sixty three patients were included in the study. The male to female ratio was 1.4:1. The mean age was 66 years for females and 61 years for males (*p* < 0.0001). A significantly larger percentage of males were underweight compared to females (60% vs. 32%, *p* < 0.001). There were no differences between the genders with regards to performance status, dysphagia grade and duration and tumor length, location and degree of differentiation. There were significant differences between risk factor exposure between the two genders. Smoking and alcohol consumption was an association in more than 70% of males but in less than 10% of females There was no difference survival. (4) Female patients with oesophageal squamous cell carcinoma (OSCC) are older and have a higher body mass index (BMI) than their male counterparts. Traditionally purported risk factors of smoking and alcohol consumption are infrequent associations with OSCC in female patients and other environmental risk factors may be more relevant in this gender.

## 1. Introduction

Oesophageal squamous cell carcinoma (OSCC) is common in Africa and has a male preponderance [1]. Some studies, however, have demonstrated parity between the genders [2]. These differences seem to coincide with differences in incidence in the area as well as differences in histological types. Areas with a tendency towards equality in terms of gender incidence are usually high incidence areas with squamous cell carcinoma as the predominant type [2,3].

Tobacco smoking and alcohol consumption are considered the two most significant environmental risk factors for oesophageal squamous cell carcinoma [4,5,6,7]. In addition the effect of the two factors combined is reported to be synergistic [8,9]. The disease has also been shown to have a relationship with patient body mass index (BMI) with an increased BMI being associated with a lower risk of OSCC [3]. Other commonly studied risk factors include specific nutrient deficiencies, exposure to polycyclic aromatic hydrocarbons, infectious agents such as human papillomavirus (HPV) and exposure to hot food and beverages [10].

Oesophageal squamous cell carcinoma is reported to have a distinctive geographical distribution, with the highest incidence occurring in the Asian oesophageal cancer belt which extends from North Central China through Central Asia to Northern Iran. The area with the second highest incidence is along the African oesophageal cancer corridor extending from Eastern to Southern Africa [11,12,13,14]. While extensive research has been performed with respect to the former, the latter high incidence area has been relatively neglected by researchers [13,14].

Gender dynamics with respect to OSCC and its associated risk factors are not fully understood. There is a paucity of information on the gender differences in OSCC, particularly in the African population. There is marked discrepancy in the gender ratio between different regions and the role of different risk factors in the different genders is not clearly defined.

In this paper we analyzed the differences between males and females with OSCC presenting to a tertiary hospital in Kwazulu-Natal, South Africa. Our objectives were to outline the main differences between the genders and to ascertain whether female patients have the same risk factors as their male counterparts.

## 2. Materials and Methods

We performed a prospective case series study of all patients diagnosed with OSCC who were referred to Grey’s Hospital, Pietermaritzburg between April 2016 and September 2019. Greys hospital is a tertiary referral hospital which provides a service to one of three areas in the province of Kwa-Zulu Natal, South Africa. The area has a population of approximately 3 million; two thirds of which live in rural areas [15,16]. We collected data on clinical characteristics, risk factor exposure, tumor pathological factors, management and outcome.

All patients with confirmed OSCC were included in the study. Patients in whom a diagnosis of OSCC could not be confirmed histologically were excluded. The purpose of the study was explained to each patient, informed consent was obtained and a proforma completed. The form included demographic details, clinical history and examination, endoscopic and pathological findings and treatment administered. Management options were described as either palliative or curative where surgical resection was the commonest curative modality while insertion of a self-expanding metal stent was the commonest palliative modality.

To determine mortality, the records of certified deaths maintained by the South African Department of Home affairs were searched by the identity number of each subject. Where death had been recorded, the date of death was noted and the survival period from time of entry into the study calculated. Survival thereafter was calculated using Kaplan–Meier curves and the effects of a number of potentially predictive variables were estimated using Cox proportional hazards regression.

Significant weight loss was defined as weight loss resulting in a change in the size of the clothing they could wear comfortably. Body mass index (BMI) was defined as normal if it was between 18.5 and 25. A BMI above 25 was classified as overweight and below 18.5 as underweight [17].

We assessed severity of dysphagia using the Mellow and Pinkas dysphagia score [18]. This is a 5-scale grading system where Grade 0 denotes the ability to swallow normally, Grade 1—dysphagia to normal solids, Grade—2 dysphagia to soft solids, Grade—3 dysphagia to liquids and solids and Grade 4—inability to swallow saliva. Additional symptoms were defined as any symptoms other than dysphagia or weight loss that are experienced by our patients.

Performance status was measured using the Eastern Cooperative Oncology Group score [19] as follows; Grade 0—fully active with no physical restrictions; Grade 1—some physical restriction but able to carry out light work; Grade 2—ambulatory and capable of self-care but unable to carry out any work activity. Confined to bed or wheelchair less than 50% of waking hours; Grade 3—capable of only limited self-care. Confined to bed or wheelchair for more than 50% of waking hours; Grade 4—completely disabled. Unable to carry out any self-care. Totally confined to bed or chair.

Endoscopic findings that were documented were tumor location and length. The tumor location was measured as distance from incisors as indicated on a standard endoscope where proximal oesophagus was defined as less than 23 cm from incisors, mid-oesophagus as 23–30 cm from incisors and distal oesophagus as greater than 30 cm from incisors. Tumor pathology was defined as well differentiated if the cells closely resembled normal oesophageal epithelium, moderately differentiated if they only resembled normal epithelium to a moderate degree and poorly differentiated if there was minimal resemblance to normal epithelium [20].

### 2.1. Statistical Analysis

Results are reported as mean (standard deviation) for normally distributed data and median (Interquartile range) for skewed data. Means and medians were compared using a one-tailed unpaired Student’s t test or Wilcoxon–Mann–Whitney U test as determined by the distributions of the variables. The distributions of categorical variables were analysed with Fisher’s exact test or chi-squared test as appropriate.

### 2.2. Ethics

The study was approved by the Biomedical Research Ethics Committee of the University of KwaZulu-Natal (UKZN), Durban, South Africa (Certificate number BF270/15).

## 3. Results

Three hundred and sixty three patients were included in the study. There were 209 males and 154 females with a male to female ratio of 1.4:1. Clinical characteristics of the patient cohort are shown in Table 1.

Men were significantly younger and there was a significantly larger number of females over the age of 70 compared to males (53 females vs. 31 males, *p* = 0.0005). The age distribution of the patients is shown in Figure 1.

There was a significant difference in BMI between males and females and a larger proportion of males were underweight (115 males vs. 45 females, *p* < 0.0001). Dysphagia and weight loss were equally distributed between the genders as presenting complaints while female patients were more likely to complain of other symptoms in addition to dysphagia and weight loss. There were no differences between the genders with regards to performance status, dysphagia grade and duration and whether the patient was from a rural or urban area (Table 1).

There were significant differences with all the risk factors investigated in the study. Table 2 shows the differences in risk factors exposure between the two sexes. Males were 10 times more likely to be past or present smokers. The average number of smoking pack years for males was more than three times that of females. Males were also 10 times more likely to drink alcohol, including traditional beer.

Tumor characteristics and management are shown in Table 3. There were no significant differences in tumor characteristics between males and females. Only ten patients (2.8%) were offered curative treatment while the rest were eligible for palliative management only and there was no difference between the two genders (*p* = 0.527). The commonest reason for patients being eligible for palliative management only was poor performance status (ECOG >1). By the end of the study, 179 (49.3%) of 363 subjects were known to have died. Kaplan–Meier survival curves are shown in Figure 2. The median length of survival noted in the group of confirmed deaths is 15 weeks and there was no difference between the genders (*p* = 0.24).

## 4. Discussion

This study reveals a number of differences between males and females with oesophageal squamous cell carcinoma (OSCC).

Oesophageal cancer is considered to be more common in males world-wide with a male to female ratio of 3:1 [21]. In sub-Saharan Africa, the reported male to female ratio is 2:1 [22]. The ratio was less than 1.5:1 in our study suggesting a relatively higher incidence among females compared to other parts of the world. This is consistent with gender ratio from other high prevalence areas in the world and supports the postulate that as the incidence of oesophageal cancer in a specified area increases, the male to female ratio approaches 1:1 [23]. Other studies from South Africa support the finding of a lower male to female ratio in oesophageal cancer patients compared to other parts of the world [24,25]. The reasons for the tendency toward parity between genders are not known.

Our female patients were significantly older than their male counterparts. Advanced age is considered a risk factor for OSCC and the fact that female patients are diagnosed at a more advanced age is also supported by data from the east [26]. The reason for the more advanced age at diagnosis among females is unknown but may be related to the difference in environmental risk factor exposure between the two genders.

The BMI in our male patients was significantly lower than that of our female patients. This difference may be due to the much greater frequency of smokers, past or present, in our male cohort, since smoking itself is associated with a lower BMI [27]. A potential confounder for this finding is that women in the general population of South Africa, including those from KwaZulu-Natal, have a significantly higher BMI than their male counterparts [28,29]. The inverse relationship between OSCC and BMI is well established, with a lower BMI indicating an increased risk as well as a poorer prognosis [30,31].

The biggest difference between genders in our study was the numbers of smokers and alcohol drinkers. Seventy nine percent of the males were past or present smokers while the number was only 8% amongst the female patients. A similarly large discrepancy exists between male past and present drinkers compared to females with oesophageal cancer. Seventy six percent of males admitted to drinking alcohol while only 10% of females did. These statistics suggest that smoking and drinking may not be significant risk factors for OSCC in females or that the exposure at least for smoking is less intense which may relate to the later age of onset seen in our female patients.

For males, the two risk factors cannot be ignored since the exposure is much higher than that of the general population of South Africa. In South Africa as a whole, 33% of males have a history of smoking (past or present), while 48% of males have a history of alcohol consumption. These numbers are much lower than those of our cohort (79% and 76% respectively). Our female patients on the other hand have an exposure similar to that of the rest of the country [29,32].

It is well known that cigarette smoking and alcohol intake increases the risk of OSCC [33] but whether the increase in risk applies to females is not clear. Studies investigating smoking and alcohol and OSCC risk usually combine males and females as well as OSCC and adenocarcinoma, making it difficult to draw conclusions on the specific relationship of gender with OSCC [34]. The association cannot be ignored since it is estimated that more than 90% of cases of OSCC can be attributed to smoking and drinking.

Findings in studies from other parts of the world differed slightly from ours. Most show a lower association of smoking and alcohol intake with OSCC in female patients but the difference between the genders is not as large and significant as in our study. Overall, the two habits accounted for between 31% and 50% of OSCC cases in women; much higher than the 10% in our study [35,36,37,38].

Tobacco smoking and alcohol intake is less prevalent among females in the general population of South Africa compared to males [29]. While this may partially explain the differences we found, it fails to explain why our female patient have such a low number of smokers and alcohol drinkers, especially considering that these two risk factors are well established in OSCC [12,13].

It is possible that smoking and alcohol as risk factors for OSCC are less important in our female population than in other series and that there are other risk factors involved that need to be quantified. This seems to be the case in other high prevalence areas as well [39,40].

Low socioeconomic status may be one of the factors playing a larger role than tobacco or alcohol in our study. Area 2, which refers to Grey’s hospital as the tertiary referral centre can be considered as having a lower socio-economic status compared to the rest of South Africa and even compared to the rest of the province. The area does in fact have a lower percentage of individuals with access to formal dwelling, flush toilets, piped water and electricity compared to rest of the province and country. In addition, the proportion of people unemployed and without any schooling is also larger than that of the rest of the province and country [16].

Other factors that have been implicated as possible risk factors for OSCC include exposure to polycyclic aromatic hydrocarbons (PAHs), thermal injury to the oesophagus, nutritional deficiencies and poor oral hygiene. Increased exposure to PAHs may occur as a result of cooking over indoor fires [41,42,43]. These factors may play a larger role in our female patients with OSCC and need to be quantified in future studies.

The lack of follow up data is the main limitation of this study. We were unable to obtain confirmation of death or survival in half the patients. These are therefore excluded from the survival analysis, which is restricted to those subjects in whom death has been confirmed. We are, therefore, unable to estimate survival with certainty, though we are able to describe our experience with those subjects who are known to have died. There does not seem to be any difference between genders in terms of outcome considering that the vast majority of patients from both groups (97%) were only eligible for palliative management. The median survival of only 15 weeks underscores the devastating prognosis that accompanies this condition.

## 5. Conclusions

The male to female ratio of oesophageal squamous cell carcinoma is lower in our patients compared to some other regions of the world. Our female patients are older and have a higher BMI than our males. In our females, smoking and alcohol consumption had a markedly lower association with OSCC than in males and the association is even lower than that of females from other studies, implying that there may be other, more important risk factors in our female patients that merit elucidation. None of the differences appear to impact on prognosis which remains poor.

## Figures and Tables

**Figure 1 ijerph-17-07086-f001:**
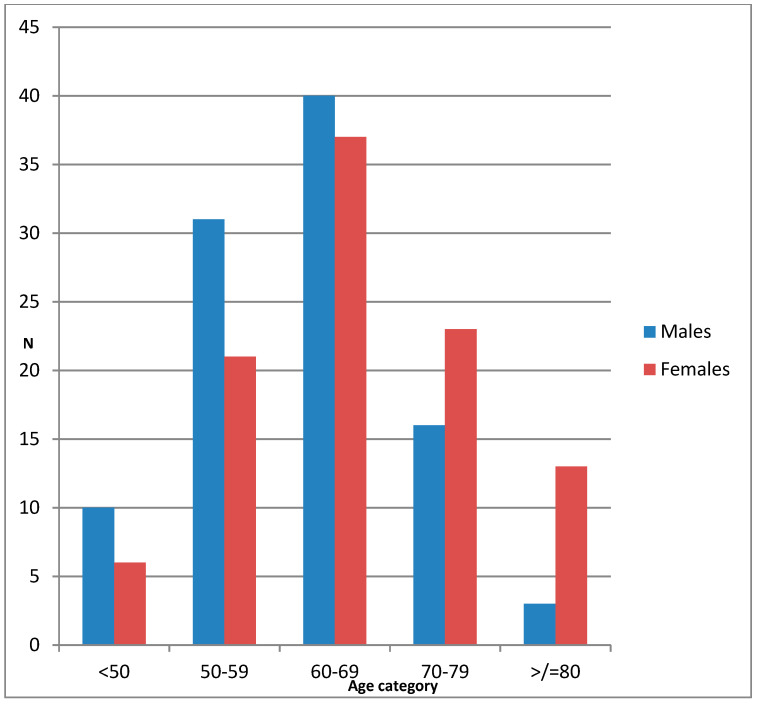
Age distribution of patients according to gender.

**Figure 2 ijerph-17-07086-f002:**
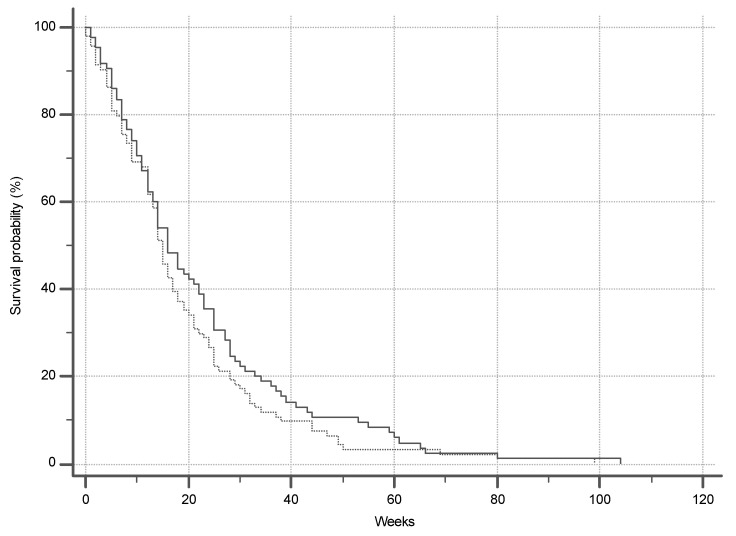
Survival in group of confirmed deaths.

**Table 1 ijerph-17-07086-t001:** Clinical characteristics of cohort.

	All		Male		Female		*p* Value
Number of Subjects	363		209		154		
	Mean	SD	Mean	SD	Mean	SD	
Age (years)	63		61.4	±9.6	66	±11.6	<0.0001
Dysphagia grade	2.2		2.2	±0.98	2.3	±0.86	0.71
Dysphagia duration (months)	4.5		4.4	±2.6	4.6	±2.4	0.65
ECOG Score ^1^	1.75		1.7	±0.89	1.8	±0.83	0.32
BMI ^2^	19.9		18.4	±3.83	21.6	±5.84	<0.001
	No	%	No	%	No	%	
Patients from urban area	104	28.8	59	28.4	45	29.4	0.91
Patients with weight loss	286	78.5	164	78	122	79	0.86
Patients with additional symptoms	75	21	30	14	45	29	0.005

^1^ ECOG—Eastern Cooperative Oncology Group; ^2^ BMI—body mass index.

**Table 2 ijerph-17-07086-t002:** Risk factors according to gender.

	All	Males	Female	*p* Value
Smoking history (%)	160 (48.3)	148 (79)	12 (8)	<0.0001
Mean pack years (SD)	15.99	16.59 (12.89)	5.14 (3.23)	0.021
Alcohol history (%)	156 (47.3)	142 (76)	14 (10)	<0.0001
Traditional beer (%)	129 (39.6)	118 (63)	11 (8)	<0.0001

**Table 3 ijerph-17-07086-t003:** Tumor characteristics and management type.

	All	Males	Females	*p* Value
Tumor location (%)				
Proximal	64 (19)	40 (20)	24 (17)	0.95
Middle	167 (49)	84 (42)	83 (59)	
Distal	111 (32)	77 (38)	34 (24)	
Tumor histology				
Well-differentiated	22 (8)	9 (5)	13 (11)	0.8
Moderately- differentiated	233 (81)	147 (85)	86 (75)	
Poorly-differentiated	32 (11)	17 (10)	15 (14)	
Mean tumor length (cm)	6.6	6.8	6.4	0.31
Patients receiving palliative management (%)	351 (97)	201 (97)	150 (98)	0.53

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
