# Peer review of "Gender Differences in Oesophageal Squamous Cell Carcinoma in a South African Tertiary Hospital"

_ijerph, 2020, doi:10.3390/ijerph17197086_

Round 1

Reviewer 1 Report

Manuscript evaluated the gender based differences in occurrence of oesophageal squamous cell carcinoma (OSCC) by utilizing data obtained from OSCC patients attending a tertiary hospital in a region of South Africa. The authors adopted a prospective study approach in collection of their result findings and they reported that while no differences exist in pathological and clinical appearance between the different gender, OSCC risk factors and BMI status constituted significant variables in which disparities exist between the males versus females. The manuscript provides interesting findings, considering the limited information about OSCC in that region of the world.

COMMENTS:

Firstly, the title of the article is too broad and their result findings cannot be extrapolated to the other African countries. Authors should rephrase their manuscript title, considering the fact that samples were collected only from South Africa (as a matter of fact, from a singular hospital in just a province of South Africa). The context “African setting” is therefore grossly out of place in the title.

Secondly the study objective is not well spelt out both in the abstract and manuscript body of text. Authors should provide a clear and distinctive objective of study for readers to fully comprehend their research purpose.

In abstract section “Female patients with OSCC are older and have a higher BMI than their male counterparts” appear twice in a similar fashion in lines 15-16 & lines 21-22. Eliminate one.

In conclusion section (line 211) which states that “…prevalent among females in our population…” is not factually backed by empirical evidence in the manuscript. While the male:female ratio may by low in study samples, that does not necessarily translate to higher prevalence in terms of total number in entire population. Authors may revisit and reframe this statement.

The significant of the BMI data in OSCC is confounded by a similar BMI trend in the general population, as authors also indicated in line 156-158. It appears therefore that the use of BMI differential between genders in OSCC cases lacks any concrete factuality.

Line 40-42 requires a reference.

Just wondering the essence of plotting the survival curve with insufficient/incomplete data, running to about half of the study sample size. This shows a poor study design since research is meant to be a prospective case series. Authors need to include and explain how patients follow up was conducted in the material & methods section, as most of the data presented appear to have been collected as a “snap-shot” of just one hospital visit.

Author Response

Response to reviewers

We thank both reviewers for their valuable comments and input. We have addressed each comment individually and made changes to the document as recommended. Where we disagreed, we stated it in our rebuttal below. The changes to the manuscript are highlighted in red.

Reviewer 1

Thank you for your comments and input. We will address each one individually.

Comment 1

Firstly, the title of the article is too broad and their result findings cannot be extrapolated to the other African countries. Authors should rephrase their manuscript title, considering the fact that samples were collected only from South Africa (as a matter of fact, from a singular hospital in just a province of South Africa). The context “African setting” is therefore grossly out of place in the title.

Response1.

We agree that the title is too broad and have changed it to ‘Gender differences in oesophageal squamous cell carcinoma in a South African tertiary hospital.’

Comment 2

Secondly the study objective is not well spelt out both in the abstract and manuscript body of text. Authors should provide a clear and distinctive objective of study for readers to fully comprehend their research purpose.

Response 2

We added a sentence to the abstract as well as the introduction of the main text specifying the objectives of the study.

Comment 3

In abstract section “Female patients with OSCC are older and have a higher BMI than their male counterparts” appear twice in a similar fashion in lines 15-16 & lines 21-22. Eliminate one.

Response 3

We have eliminated the line in the results section and re-worded the results.

Comment 4

In conclusion section (line 211) which states that “…prevalent among females in our population…” is not factually backed by empirical evidence in the manuscript. While the male:female ratio may by low in study samples, that does not necessarily translate to higher prevalence in terms of total number in entire population. Authors may revisit and reframe this statement.

Response 4

We agree and have changed the sentence to ‘The male to female ratio of oesophageal squamous cell carcinoma is lower in our patients compared to some other regions of the world.’ This statement is backed up by our data and references 21 and 22.

Comment 5

The significant of the BMI data in OSCC is confounded by a similar BMI trend in the general population, as authors also indicated in line 156-158. It appears therefore that the use of BMI differential between genders in OSCC cases lacks any concrete factuality.

Response 5

We agree that the BMI trend is similar to the general population. That is why we argue that it may explain our finding. This cannot be stated as a certainty though, and there may be other reasons for the difference in BMI. The difference in the incidence of smoking and alcohol consumption may be one such reason. There is much uncertainty around this issue and we feel therefore that it is relevant for inclusion in the discussion.

Comment 6

Line 40-42 requires a reference.

Response 6

The lines referred to on our template does have references . They are stated below

‘Other commonly studied risk factors include specific nutrient deficiencies, exposure to polycyclic aromatic hydrocarbons, infectious agents such as HPV and exposure to hot food and beverages10.

Oesophageal squamous cell carcinoma (OSCC) is reported to have a distinctive geographical distribution, with the highest incidence occurring in the Asian oesophageal cancer belt which extends from North Central China through Central Asia to Northern Iran. The area with the second highest incidence is along the African oesophageal cancer corridor extending from Eastern to Southern Africa11-14.’

Comment 7

Just wondering the essence of plotting the survival curve with insufficient/incomplete data, running to about half of the study sample size. This shows a poor study design since research is meant to be a prospective case series. Authors need to include and explain how patients follow up was conducted in the material & methods section, as most of the data presented appear to have been collected as a “snap-shot” of just one hospital visit.

Response 7

We agree that this is a limitation of the study. We have included the follow up in the methods section. See below

‘To determine mortality, the records of certified deaths maintained by the South African Department of Home affairs were searched by the identity number of each subject. Where death had been recorded, the date of death was noted and the survival period from time of entry into the study calculated.’

The fact that this is not an accurate reflection of survival was stated in the discussion section. See below

‘The lack of follow up data is the main limitation of this study. We were unable to obtain confirmation of death or survival in half the patients. These are therefore excluded from the survival analysis, which is restricted to those subjects in whom death has been confirmed. We are therefore unable to estimate survival with certainty, though we are able to describe our experience with those subjects who are known to have died. There does not seem to be any difference between genders in terms of outcome considering that the vast majority of patients from both groups (97%) were only eligible for palliative management.’

Despite the limitation, it provides valuable information in that the differences observed did not translate into differences in survival.

Reviewer 2 Report

Dear authors

Attached you can find your manuscript with my suggestions and corrections.

The abstract can be significantly improved.

Along the manuscript highlighted in yellow or with remarks, what I think it can be improved.

The inclusions and exclusions criteria must be clearly started at the material&methods chapter.

Good work

Keep safe.

Author Response

Response to reviewers

We thank both reviewers for their valuable comments and input. We have addressed each comment individually and made changes to the document as recommended. Where we disagreed, we stated it in our rebuttal below. The changes to the manuscript are highlighted in red.

Reviewer 2

Thank you for your comments and input. We addressed each comment individually below as well as in the text boxes provided on the pdf document. Please see the attached PDF file

  1. Line 11 - We added a sentence on our aim for the study
  2. Line 19 - We deleted the p value as suggested
  3. Line 38 - Human papilloma virus written in full
  4. Line 50 - We added aim and objectives in the introduction
  5. Line 72 - Body mass index written in full
  6. Line 109 and 115 - vs changed to italic
  7. Line 122 - Sentence was changed to ‘Males were 10 times more likely to be past or present smokers. The average number of smoking pack years for males was more than three times that of females.’ in order to clarify.
  8. line 126. That is correct. The low number of patients that can be offered curative treatment is related to the late presentation
  9. Line 135. We agree that there are no statistically significant differences between any of the variables described on the table. We still feel that it is relevant though since it points out important negative findings. The variables described may all have an impact on outcome in patients with oesophageal cancer.
  10. Line 146. The reasons we stated were speculation and therefore there are no references. We deleted the speculative portion of the sentence.
  11. Line 165. We agree that tobacco smoking and alcohol consumption are listed as common risk factors for oesophageal cancer in the literature. Our data challenges this statement when it comes to females and a sample size of 363 patients with 209 males and 154 females with the size of the discrepancy we reported is enough to be significant. In fact the difference between the two groups was so large that the p value was < 0.000000001 for both risk factors. We therefore disagree that our findings are related to sample size. We feel that our finding is adding to the literature on this subject as it refutes the fact that these two risk factors play a major role in female patients from our area with this disease. It is one of the main points of the article. We hope to stimulate more research on the topic and we ourselves will continue to study this and present data from larger sample sizes in the future.
  12. Line 183. The sentence has been changed to ‘Tobacco smoking and alcohol intake is less prevalent among females in the general population of South Africa compared to males’ for clarity as suggested.
  13. Line 198. We agree with the reviewer and will keep in mind for the follow up study

Round 2

Reviewer 1 Report

Authors have made significant changes to improve their manuscript.

First sentence of abstract ought to be written as "Oesophageal squamous cell carcinoma is common in Africa and has a male preponderance".